# Low precipitation due to climate change consistently reduces multifunctionality of urban grasslands in mesocosms

Sandra Rojas-Botero[1]*, Leonardo H. Teixeira[1], Johannes Kollmann[1,2]

**1** Chair of Restoration Ecology, TUM School of Life Sciences, Technical University of Munich, Freising, Germany, **2** Norwegian Institute of Bioeconomy Research (NIBIO), Ås, Norway

☉ These authors contributed equally to this work.
* sandra.rojas-botero@tum.de

**Data Availability Statement:** The DOI to access the data in the mediaTUM repository is https://doi.org/10.14459/2022MP1691146.

**Funding:** SR-B was supported by the joint project LandKlif, funded by the Bavarian Ministry of

## Abstract

Urban grasslands are crucial for biodiversity and ecosystem services in cities, while little is known about their multifunctionality under climate change. Thus, we investigated the effects of simulated climate change, i.e., increased $[CO_2]$ and temperature, and reduced precipitation, on individual functions and overall multifunctionality in mesocosm grasslands sown with forbs and grasses in four different proportions aiming at mimicking road verge grassland patches. Climate change scenarios RCP2.6 (control) and RCP8.5 (worst-case) were simulated in *walk-in* climate chambers of an ecotron facility, and watering was manipulated for normal vs. reduced precipitation. We measured eight indicator variables of ecosystem functions based on below- and aboveground characteristics. The young grassland communities responded to higher $[CO_2]$ and warmer conditions with increased vegetation cover, height, flower production, and soil respiration. Lower precipitation affected carbon cycling in the ecosystem by reducing biomass production and soil respiration. In turn, the water regulation capacity of the grasslands depended on precipitation interacting with climate change scenario, given the enhanced water efficiency resulting from increased $[CO_2]$ under RCP8.5. Multifunctionality was negatively affected by reduced precipitation, especially under RCP2.6. Trade-offs arose among single functions that performed best in either grass- or forb-dominated grasslands. Grasslands with an even ratio of plant functional types coped better with climate change and thus are good options for increasing the benefits of urban green infrastructure. Overall, the study provides experimental evidence of the effects of climate change on the functionality of urban ecosystems. Designing the composition of urban grasslands based on ecological theory may increase their resilience to global change.

## Introduction

Urban green infrastructure supports biodiversity and ecosystem services since it serves as habitat, regulates temperature and stormwater, improves air and water quality, and increases aesthetics [1]. With increased anthropogenic $[CO_2]$, higher temperatures, changed precipitation,

Science and the Arts via the Bavarian Climate Research Network (bayklif; F.7-F5121.14.2.3/14/9); DFG (INST 95/1184-FUGG) supported the establishment TUMmesa. https://www.stmwk. bayern.de/ https://www.dfg.de/ The funders had no role in study design, data collection and analysis, decision to publish, or preparation of the manuscript.

**Competing interests:** The authors have declared that no competing interests exist.

and more extreme events [2], functional urban ecosystems are relevant for mitigating the impacts of climate change via ecosystem services [3–5]. At the same time, these ecosystems are also threatened by climatic extremes [6, 7]. The various components of climate change can alter the physiological responses of organisms, the dynamics of populations, and the development of communities, ultimately affecting ecosystem functions and services [8]. In grasslands, for example, higher temperature increases ecosystem respiration and reduces productivity [9], while elevated $[CO_2]$ increases soil respiration [10] and aboveground productivity [11]. In turn, water stress affects net ecosystem $CO_2$ exchange [12, 13] and increases soil water retention, nutrient leaching, and soil fertility [13].

Grasslands represent a high proportion of urban ecosystems [14] subjected to strong human influence determining their composition and functioning (e.g., light pollution, soil pollutants, trampling, high levels of habitat fragmentation, soil compaction, etc.; [15–18]). Urban grasslands range from relictual patches remaining from extensively managed agricultural grasslands [17, 19] up to novel grasslands arising in brownfields and strictly designed ones [19, 20]. Among those, urban lawns are the most frequent components of green infrastructure [21]. However, urban lawns have become less diverse due to regular mowing, fertilization, and irrigation [22] and altered biogeochemical cycles [23, 24]. With growing initiatives to replace intensively managed lawns in cities with diverse meadows [19, 25], understanding the functioning and resilience of the latter to climate change will allow for better design and more effective design and restoration of green urban spaces. If adequately implemented, urban grasslands can harbor large numbers of plant species [26], offer habitat and food for animal species [27, 28], and provide numerous ecosystem services to humans [1].

A deeper understanding of the biotic drivers of the multiple functions performed by urban grasslands and their responses to climate change helps identify synergies and trade-offs among ecosystem functions, with marked consequences to ecosystem services [29]. During the past decades, various studies highlighted the role of taxonomic and functional diversity on single ecosystem functions and ecosystem multifunctionality [29, 30], particularly in experimental and semi-natural grasslands [31]. Besides species richness, functional diversity and functional type composition explain ecosystem functioning in plant communities [32, 33]. In grasslands, for example, grasses and forbs are useful surrogates of functional diversity, depicting specific adaptations to the physical environment, patterns of resource use, and resilience to disturbances [33, 34], also under climate change [35]. Therefore, variation in the abundance of grasses and forbs might translate into changes in grassland functioning.

Despite increasing evidence that community composition affects the resilience of grassland functioning to climate change, most studies focus only on species richness as the primary driver of resilience to climate change (but see [36]). Moreover, few studies consider the combination of different climate-change components (e.g., [37, 38]). Thus, the effects of climate change on varying compositions of grasslands remain poorly understood [39] and hamper the recognition of relevant aspects to the design and management of multifunctional urban grasslands.

Here, we experimentally evaluate whether the simultaneous impacts of simulated climate change and functional type composition affect the multifunctionality of urban grasslands using mesocosms in an ecotron facility. To address current research gaps, we assess the combined effects of three components of climate change, i.e., higher $[CO_2]$, increased temperatures, and reduced precipitation, on the multifunctionality of experimental grasslands composed of contrasting proportions of forbs vs. grasses. We expected: (i) higher temperature and $[CO_2]$ to enhance carbon cycling of mesocosm grasslands; (ii) reduced precipitation to decrease vegetation performance and ecosystem multifunctionality; (iii) differing functional type composition to produce contrasting effects on grassland functioning under current

environmental conditions, with climate change enhancing specific responses of grass- and forb-dominated communities; and (iv) mesocosm grasslands with high evenness of plant functional types to exhibit higher multifunctionality under climate change.

## Materials and methods

We implemented a mesocosm experiment imitating early phases of grassland establishment in urban settings, assuming the sowing of target species in prepared bare soil of urban road verges (S1 File). In this experiment, we manipulated the relative abundance of forbs and grasses to assess grassland functioning in response to variations in plant functional types and simulated climate change (decreased precipitation, increased temperature, and [$CO_2$]). We measured different indicator variables to assess the functioning of grasslands: vegetation structure and productivity, soil, and water regulation as proxies of functions related to ecosystem services expected from multifunctional urban ecosystems.

### Design of experimental grassland communities

We tested four grassland mixtures: A seed mixture of only grasses, representative of species-poor lawns. Thereupon, the communities were complemented by a mix of native forbs used in restored urban road verges [40] to establish four experimental mixtures of forbs and grasses, respectively, i.e., 0:100 (F0), 50:50 (F50), 75:25 (F75), and 100:0 (F100). The mixtures were selected to represent two extremes of the gradient (i.e., F0 and F100), a commonly used cheap 1:1 combination of forbs and grasses (F50), and a mixture that prioritizes forbs for the so-called 'pollinator-friendly meadows' (F75). Moreover, since too much grass biomass negatively affects the target species in bioengineered grasslands for road verges [41], we decided against testing a 75% grass mixture.

We selected five native grass and 26 forb species (five being legumes) occurring in urban habitats of Bavaria, following criteria relevant to urban pollinators and plant performance in urban contexts (S1 Table contains species information). Mixture F0 comprised five grass species (one plant family), F100 contained included 26 species (12 families), and F50 and F75 had 31 species (13 families). Seed mixtures were assembled by weight to a desired density of 4400 seeds m$^{-2}$. The contribution of each species to the seed mixture was adjusted to the treatments, whereby each species of the respective functional type had the same proportion (S2 Table for the detailed composition of the communities). The actual establishment of the experimental communities was not recorded, but it looked similar to the respectively prepared mixtures. Because of focusing on the influence of the plant functional types (grasses vs. forbs), species numbers were not accounted for, but considered minor when assessing the effects of functional composition on the grasslands.

### Experimental design and monitoring

To test the effects of plant functional types proportions ('forb proportion' henceforth) and climate change on urban grassland multifunctionality, we established a 10-week experiment in four walk-in climate chambers (2.4 x 3.2 x 2.2 m$^3$; area 7.7 m$^2$; S1 Fig) of the ecotron facility TUMmesa of the Technical University of Munich (Freising, Germany) [42]. We simulated environmental conditions of two IPCC climate change scenarios (RCP2.6 and RCP8.5; two chambers per scenario) reflecting environmental parameters of May–July when vegetation development is fast. For RCP2.6 (control), with temperature and [$CO_2$] similar to current values, we reproduced median hourly values of temperature between 2000 and 2019 [43] and [$CO_2$] recorded in central areas of Munich with a TDLAS system [44]. For RCP8.5 (worst-case scenario), median values of the record warm years for C Europe 2003, 2015, and 2018 were

used, while the values of $[CO_2]$ were set to double the current ones. Simulation of the worst-case climate change scenario (RCP8.5) increased $[CO_2]$ and air temperature by 486 ppm (+99.7%) and 3.1°C (+17.6%), respectively. The relative humidity was lower under RCP8.5 (–4.7%). Thus, the realized simulation of temperature and $[CO_2]$ lay within the expectations under RCP8.5 by the end of the century [2] and showed clear differentiation between the two scenarios manipulated in the climate chambers (S2 Fig). Furthermore, we chose RCP8.5 because enhanced effects of climate change are expected in urban areas [45], and recent reports on the course of climate change show that carbon emissions remain close to the track marked by this scenario [46].

Sixty-four plastic mesocosms (70 x 40 x 23 cm$^3$ (L x W x H)) were filled with a mixture of washed sand:potting substrate (70:30) in the lowest 10 cm, and urban lawn substrate in the upper 10 cm, to mimic the soil depth conditions of urban road verges (Fig 1). Each mesocosm had six 1-cm diameter holes in the bottom to allow water drainage, and a plastic container placed below to collect the drained water. Mesocosms were distributed on floodable tables (four per table). Four tables were allocated within each climate chamber. A grassland mixture (F0, F50, F75, or F100) was randomly assigned to each mesocosm on a table, while the two simulated precipitation regimes were randomized among the tables in each chamber. Thus, each mesocosm represented a combination of seed mixture, watering regime, and climate change scenario ($[CO_2]$ and temperature) (S3 Fig).

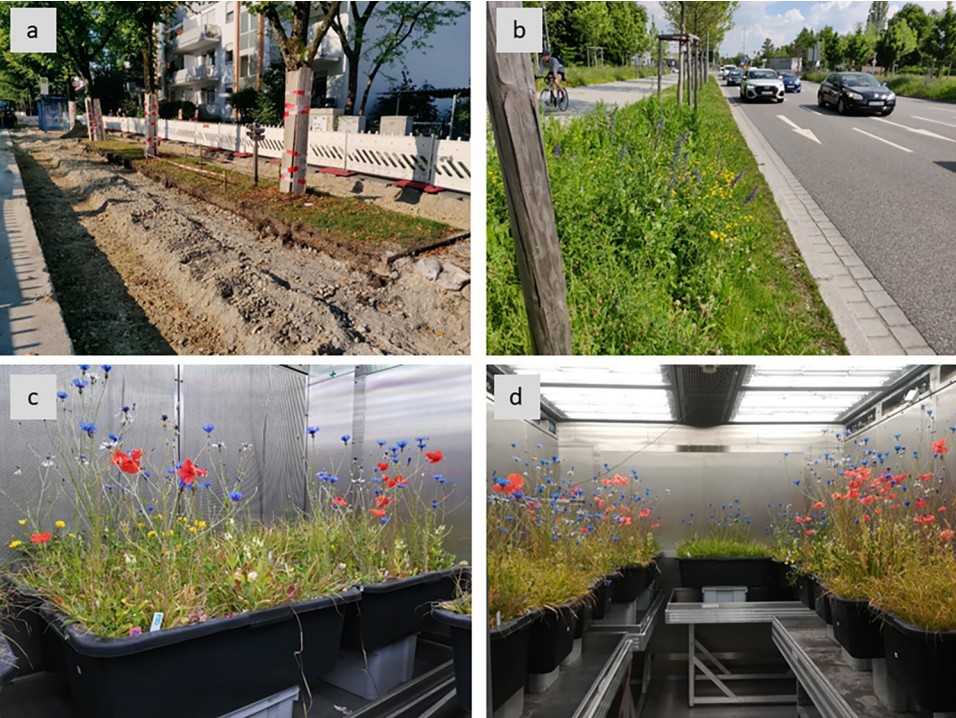

**Fig 1. Urban road verge grasslands mimicked in mesocosms in climate chamber experiment.** (a) Road verge grasslands often occur on heavily sealed ground with shallow soil and using few grass species. (b) Restored grasslands in road verges are species-rich and adapted to urban conditions to improve ecosystem services delivered. (c) In the mesocosm experiment, forb species used for restoring urban road verges were sown, and the communities developed under similar conditions (e.g., substrate characteristics, limited soil depth, and surrounding sealed space). (d) Mesocosm grasslands of varying functional compositions were submitted to two climate change scenarios in the climate chambers of an ecotron facility. Pictures of road grasslands (a) and (b) were taken in Munich, Germany (courtesy of Simon Dietzel); pictures (c) and (d) were taken at weeks 9 and 10 of mesocosm development in chambers submitted to RCP8.5 and RCP2.6, respectively. For a better overview of the communities' development under the two tested scenarios, see S2 Fig.

Each seed mixture was thoroughly spread by hand onto the prepared substrate of the mesocosms, which were kept well-watered during the first 27 days after sowing. Precipitation was simulated by watering the mesocosms from above every other day with a hose sprayer. We adopted a 20-year monthly precipitation time series for late spring and summer in Munich, i.e., the mean monthly precipitation recorded from May to July between 2000 and 2019 [47], which ranged from 101 to 120 mm and distributed over the time of the experiment considering the area of the mesocosms. For the reduced precipitation treatment, we simulated a dry period of the growing season by halving the amount of water in the mesocosms, matching the mean monthly precipitation amounts recorded in the same months in the extremely dry years 2003, 2015, and 2018. There was no fertilization during the experiment.

We measured indicator variables of ecosystem functions of interest for urban greening at each mesocosm. The measurements represented above- and belowground compartments of urban grasslands, with which we approximate ecosystem functions [31, 48–50]. A comprehensive overview of the indicator variables, experimental measurement procedures, time points and frequency with which variables were recorded, their role in urban ecosystem services, and potential responses to the manipulated components of climate change is available in S3 Table. We recorded five variables related to vegetation (i.e., above- and belowground biomass, flower production, plant cover, and plant height), constituting a good proxy for productivity and determining to a significant extent ecosystem functioning [51]. Moreover, we considered cover and height as structural components of vegetation that, besides productivity [51], relate to aboveground space filling, habitat offer, and soil protection [31, 51]. We also measured soil respiration as it correlates to C exchange and represents the largest component of ecosystem respiration [52]. Finally, using a gravimetric approach, we assessed water retention in the communities (plant-soil system), and water loss through evapotranspiration (ET) following a watering and weighing protocol adopted from studies on green roofs [53, 54], which is a proxy for rainwater regulation functions of urban grasslands in shallow road verges. For ideal storm-water regulation and transpirational cooling support, urban grasslands need to use high amounts of water when available after a rain event, and persist during periods of water scarcity [53, 54].

No permissions were necessary for conducting the mesocosm experiment since it took place in a controlled facility (TUMmesa ecotron) using native plant species produced by a certified local seed producer. Additionally, no permission was necessary to access any field sites, because this work did not involve field work.

## Data analyses

Before assessing single indicator variables, above- and belowground biomass and floral abundance were calculated per area. We averaged measures taken multiple times in each mesocosm (e.g., soil respiration), so every variable was included only once in the analyses. Water retention and loss were expressed as retained or lost water fractions (values between 0 and 1).

We calculated two indices of ecosystem multifunctionality with the eight indicator measurements: the 'averaging' and 'single threshold approaches', using the protocol of [29, 30], and implemented both with the R package 'Multifunc' version 0.8.0 [30]. Data exploration for collinearity resulted in a strong correlation between aboveground biomass and floral density (above |0.7|; S4 Fig; [55]). Therefore, we calculated each index twice, once without aboveground biomass and another without floral density. Because floral density and aboveground biomass are essential indicators of the expected functioning of urban grasslands, we calculated two values per multifunctionality approach instead of discarding one of these indicator variables. Hierarchical clustering following [29] was used before calculating the two approaches to

multifunctionality to balance calculations among correlated functions by identifying subsets of related variables (S5 Fig). This procedure prevented closed-related variables from disproportionally driving multifunctionality to certain aspects of ecosystem function [29]. After clustering the variables, an optimal number of clusters was determined following the 'elbow method', and the variables within each cluster were weighed equally by giving a proportional weight summing up to one for each cluster. Three clusters were determined, and the weighted indicators variables contributed to each multifunctionality calculation.

In the averaging approach, all standardized values of the weighed indicator variables were averaged per mesocosm. This measure of multifunctionality was modeled later against climate change components and forb proportion. In the threshold approach, we selected a 70% threshold as a realistic level of multifunctionality desired for urban road verge grasslands with high responsiveness to management [56], also tested as plausible for soil multifunctionality under conditions experimented in urban areas [57], and not overoptimistic for early stages of restoration [58]. To obtain the maximum observed value of each assessed indicator variable [30], we averaged the six highest observations per indicator variable (top 10% values) to reduce outlier effects [30, 59] and to reflect conservative maximum multifunctionality attained by the urban grassland systems we studied in the mesocosms. In the calculation, each ecosystem function exceeding 70% of the standardized maximum contributed to the multifunctionality score with its respective weighted value obtained after clustering. In light of the challenges imposed by climate change to the establishment of functional urban ecosystems, we considered this threshold to be a reasonable assumption desired for these systems. Finally, because of each approach's known advantages and disadvantages in measuring multifunctionality [30], we presented and discussed the results based on the assessment of individual measurements.

We tested the effects of forb proportion, precipitation, and RCP scenario on single indicator variables and the two metrics of multifunctionality calculated for the experimental grasslands. We considered full models with all possible two-way interactions and conducted model simplification until at least the main fixed effects remained in the model. All possible models were compared via the dredge function ('MuMIn'; [60]), and we selected the simplest ones among those with a difference in AICc < 2. For continuous data (i.e., aboveground biomass, belowground biomass, plant cover, plant height, respiration, and the multifunctionality indices), linear mixed-effects models were used ('lme4'; [61]). Data stemming from the measurement of water regulation (i.e., water retention and loss) were analyzed using beta regression ('glmmTMB'; [62]). For the floral density, we fitted a zero-inflated negative binomial mixed-effects model (ZIP) after testing and detecting an excess of zeros in the data using a simulation approach and comparing the proportion of zeros in the data to the simulated ones expected based on a Poisson model [63]. We used generalized linear mixed-effects models because they provide a flexible approach for the analysis of non-normally distributed data (i.e., counts, fractions) and the crossed character of the data in the experimental design. In all cases, we included the identity of climate chambers as a random effect. The fit of the models was checked with the DHARMa package [63]. Post-hoc analyses were applied to compare functioning between levels of forb proportion whenever a significant effect was found and to correct for multiple comparisons [64]. Data analysis was conducted with R version 4.1.2 [65].

## Results

### Response of single functions

The proportion of forbs and the simulated precipitation additively influenced aboveground biomass (Table 1; Fig 1A). Communities with forbs produced more aboveground biomass across environmental conditions than grass-only communities (F100, t = 5.02, p < 0.001; F75,

**Table 1. Effects of forb proportion, precipitation, RCP climate change scenarios, and their interactions on eight indicator variables and multifunctionality of mesocosm grasslands in the climate chambers of the TUMmesa ecotron.**

| Indicator variable | Forb proportion | | RCP scenario | | Precipitation | | RCPscen:Precip | | RCPscen:Forb prop | | Precip:Forb Prop | |
|---|---|---|---|---|---|---|---|---|---|---|---|---|
| | Statistic | p-value | Statistic | p-value | Statistic | p-value | Statistic | p-value | Statistic | p-value | Statistic | p-value |
| Above-ground biomass | 28.82 | **<0.001** | 1.49 | 0.22 | 14.64 | **<0.001** | | | | | | |
| Below-ground biomass | 58.76 | **<0.001** | 0.99 | 0.32 | 3.85 | **0.05** | | | | | | |
| Plant cover | 12.87 | **0.005** | 13.90 | **<0.001** | 47.84 | **<0.001** | | | | | | |
| Plant height | 6.38 | 0.09 | 6.68 | **0.010** | 1.45 | 0.23 | 60.56 | **<0.001** | 27.04 | **<0.001** | 16.66 | **0.001** |
| Floral density | 179.82 | **<0.001** | 6.69 | **0.010** | 17.99 | **<0.001** | | | | | | |
| Soil respiration | 4.63 | 0.20 | 61.81 | **<0.001** | 18.83 | **<0.001** | | | | | | |
| Water retention | 0.44 | 0.93 | 0.57 | 0.45 | 29.02 | **<0.001** | 8.02 | **0.005** | | | | |
| Water loss (ET) | 4.09 | 0.25 | 1.72 | 0.19 | 26.28 | **<0.001** | 6.84 | **0.009** | | | | |
| Avg. Multifunctionality-AGB | 2.07 | 0.56 | 0.08 | 0.78 | 9.44 | **0.002** | 4.25 | **0.039** | 12.79 | **0.005** | | |
| Avg. Multifunctionality-Flowers | 1.05 | 0.79 | 0.11 | 0.74 | 10.22 | **0.001** | 5.35 | **0.021** | 11.52 | **0.009** | | |
| 70% Multifunctionality-AGB | 2.19 | 0.53 | 6.77 | **0.009** | 14.75 | **<0.001** | | | | | | |
| 70% Multifinctionality-Flowers | 3.15 | 0.37 | 12.17 | **<0.001** | 14.82 | **<0.001** | 9.92 | **0.002** | | | | |

Responses were analyzed with generalized linear mixed-effects models (GLMMs). Climate chambers were considered as random factors. Shown are outputs of ANOVA tables displaying the significance of the tested drivers after model simplification. Degrees of freedom are forb proportion (df = 3), precipitation (df = 1), RCP scenario (df = 1), RCP scenario x precipitation, forb proportion x RCP scenario (df = 3), and forb proportion x precipitation (df = 3). Significance level set at $p \leq 0.05$ and depicted in bold (n = 4).

t = 3.49, p < 0.001; F50, t = 4.09, p < 0.001; complete model outputs in S4 Table). Reduced precipitation negatively influenced aboveground biomass production in all communities (t = -3.83, p < 0.001; S4 Table), regardless of climate change scenario. Even though aboveground biomass was higher under warmer temperatures and higher [$CO_2$], no statistical evidence was observed for an effect of RCP scenario (Table 1; S4 Table). Belowground biomass was additively affected by forb proportion and precipitation (Table 1, Fig 1B). Opposite to aboveground biomass, grass-only communities (F0) presented the highest root biomass under all climatic conditions (S4 Table). Communities with even functional composition (F50) had higher belowground biomass than forb-only communities (t = 2.93, p = 0.025; S5 Table). Reduced precipitation negatively affected root biomass production in all experimental communities (t = -1.96, p = 0.055).

Floral density significantly responded to forb proportion, precipitation, and RCP scenario (Table 1, Fig 1C). Density of flowers increased with forb proportion (t = [10.11, 12.54], p < 0.001), more flowers were produced under normal precipitation (t = 4.24, p < 0.001), and scenario RCP8.5 (t = 2.59, p = 0.009). Plant cover was additively influenced by forb proportion, precipitation, and climate change scenario (Table 1, Fig 1D). Plant cover increased with lower forb proportion (t = [2.07, 2.55], p < 0.05; S4 Table), and under RCP8.5 (t = 3.73, p = 0.06), but decreased with reduced precipitation (t = -6.92, p < 0.001). Forb proportion, precipitation, climate change scenario, and all two-way interactions affected mean plant height (Table 1, S4 Table, Fig 1E). All communities containing forbs were taller than grass-only communities under normal precipitation conditions, while there were no clear differences under reduced precipitation (S6 Table). Plant height was negatively affected by the interaction of reduced precipitation and scenario RCP8.5 (t = 7.78, p < 0.001), most likely due to insufficient water supply for stimulating growth, even though higher [$CO_2$] increases the fixation of C and plant growth. Under RCP8.5, grass-only communities were shorter than all other functional compositions (S6 Table), and grasslands with an even composition of forbs and grasses (F50) were significantly taller than forb-dominated grasslands (i.e., F75 and F100). Climate change

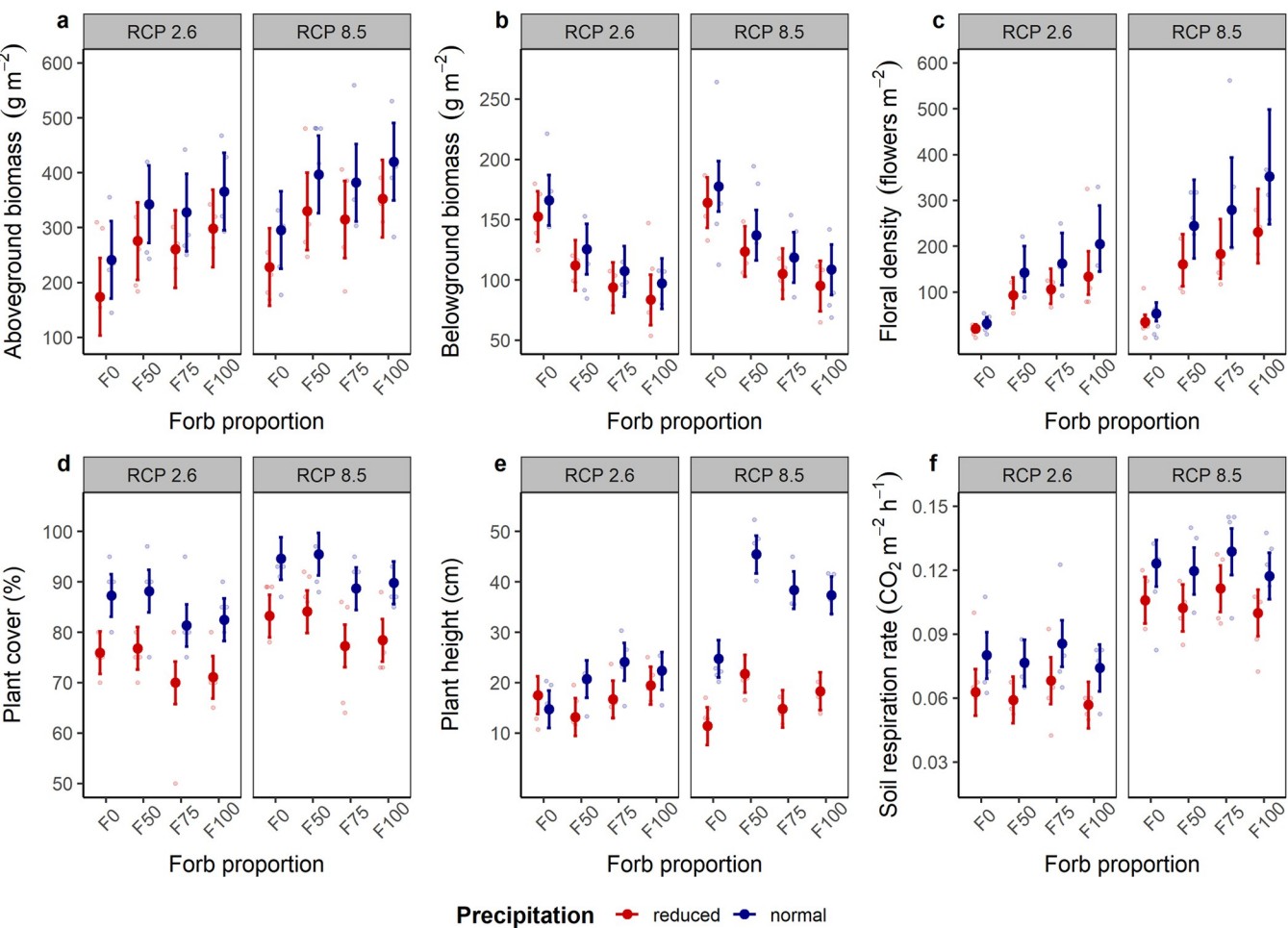

**Fig 2. Effects of forb proportion, precipitation, and climate change scenarios on indicators of grassland functioning.** (a) Aboveground biomass, (b) belowground biomass, (c) floral abundance, (d) vegetation height, (e) cover, and (f) soil respiration rate (means ± 95% confidence intervals) of mesocosm communities in the climate chambers of an ecotron facility. Data points are light-colored circles (n = 4).

scenario and precipitation additively affected soil respiration (Table 1). Soil respiration rate was lower in mesocosm grasslands with reduced precipitation (t = -4.34, p < 0.001), and under scenario RCP8.5, soil respiration increased (t = 7.86, p = 0.016; Fig 2E). Water regulation did not respond to forb proportion and RCP scenario (Table 1). In turn, both indicator variables of water regulation were affected by the interaction between precipitation and RCP scenario (Table 1). Under RCP2.6, water retention substantially increased in grasslands with reduced precipitation, while no differences in retained water fraction were found under scenario RCP8.5 with regard to the amount of precipitation the communities received. Similarly, water loss was significantly lower in communities receiving reduced precipitation under RCP2.6, whereas the fraction of water loss did not differ when precipitation was reduced under RCP8.5 (Fig 3B; S4 Table).

## Urban grassland multifunctionality

Precipitation and its interaction with RCP scenario affected averaged multifunctionality (Table 1). Decreased precipitation was overall detrimental for averaged multifunctionality,

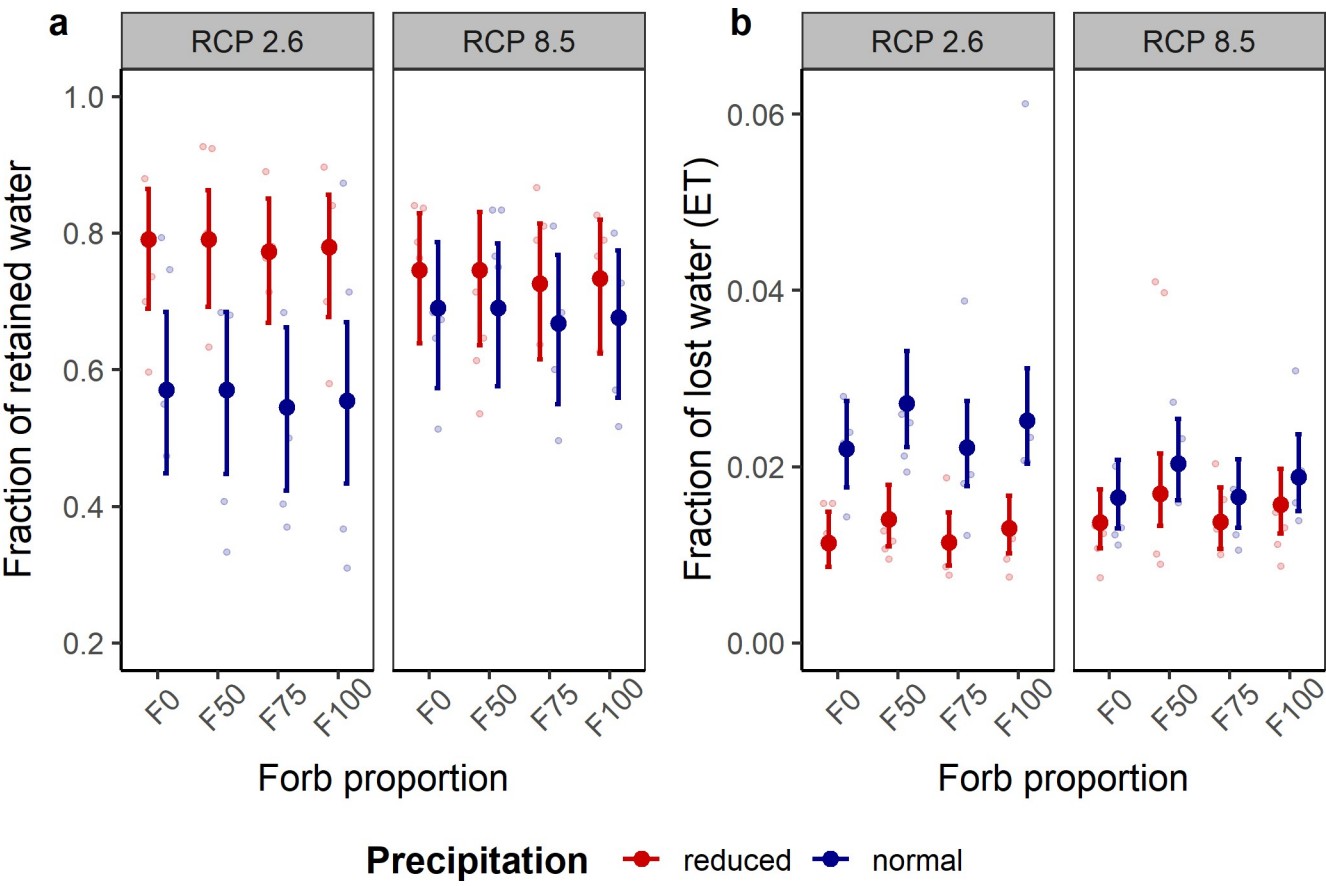

**Fig 3. Effects of forb proportion, precipitation and climate change scenarios on water regulation in grasslands.** (a) Fraction of retained, and (b) lost water by evapotranspiration [ET] (means ± 95% confidence intervals) of mesocosm communities in the climate chambers of the ecotron facility. A heavy-rain event was simulated to derive information on water regulation function. Data points are light-colored circles (n = 4).

whereas RCP8.5 in interaction with normal precipitation was beneficial for averaged grassland multifunctionality (S7 Table, Fig 4A and 4C). Under RCP8.5, grasslands evenly composed of grasses and forbs (i.e., F50) presented higher averaged multifunctionality than grass-only communities in both cases, i.e., considering the calculations using above-ground biomass and floral density (t = 3.40 and 3.29, p = 0.001 and 0.002, respectively; S7 Table). F75, in interaction with RCP8.5, positively affected multifunctionality when floral abundance was used instead of aboveground biomass for multifunctionality calculation (t = 2.05, p = 0.045), underscoring the biotic control of multifunctionality under climate change.

Only precipitation affected the functioning of mesocosm grasslands at or above a threshold of 70% multifunctionality (Table 1). We observed more ecosystem functions above this threshold under normal precipitation (Fig 4B, 4D, S7 Table). Moreover, when multifunctionality at 70% threshold considered flower density instead of aboveground biomass, the interaction between precipitation and RCP scenario became significant, with a negative effect of RCP8.5 interacting with reduced precipitation (S7 Table; Fig 4D). At lower levels of multifunctionality (e.g., 50% threshold), decreases in precipitation were consistently detrimental. At the same time, functional composition in interaction with RCP scenario remained important for increased multifunctionality, particularly in communities with even compositions of forbs and grasses, i.e., F50 (S2 File).

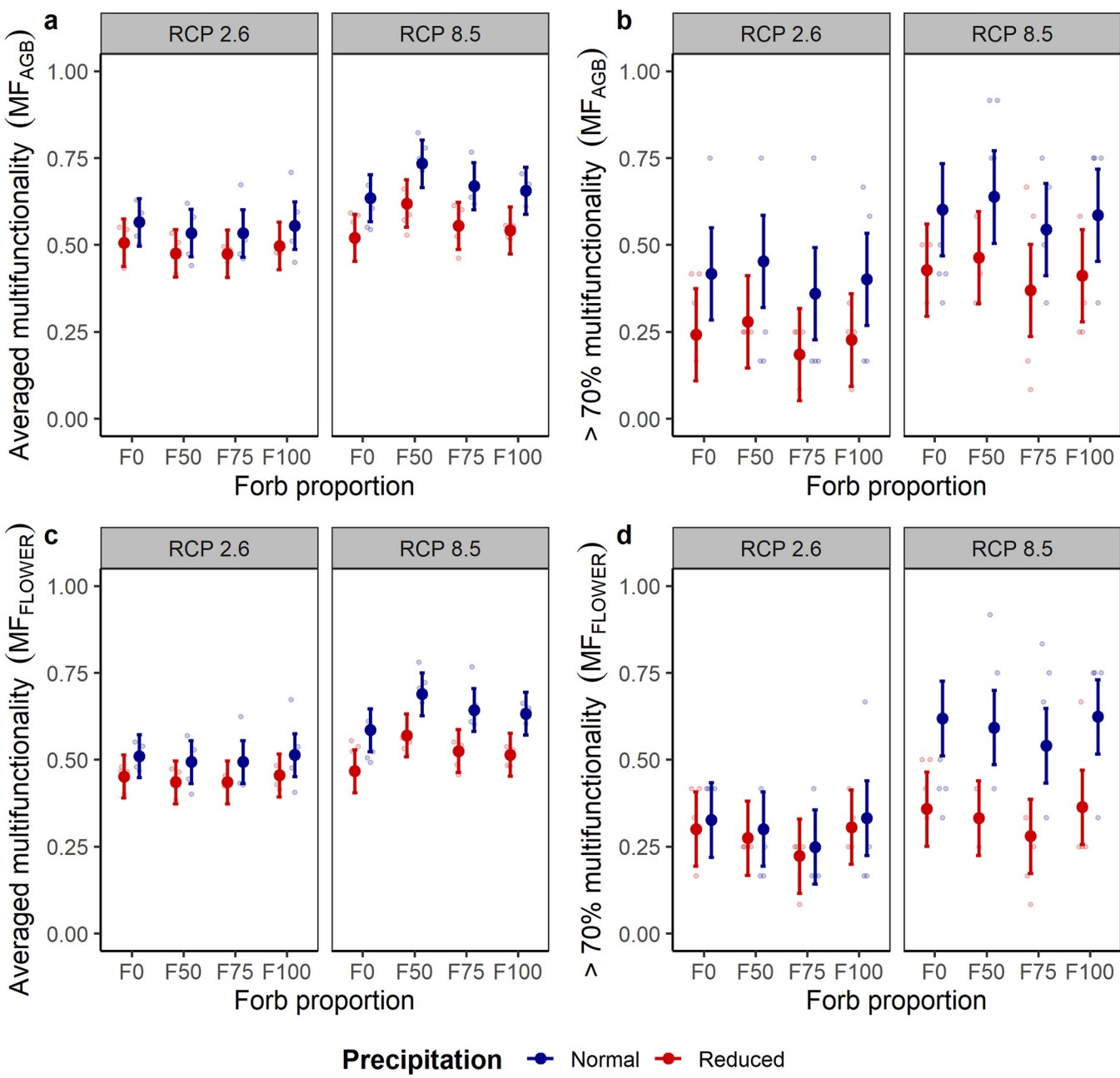

**Fig 4. Effects of forb proportion, precipitation and climate change scenario on grassland multifunctionality.** Two approaches to multifunctionality (MF) were calculated using eight indicator variables, including either aboveground biomass (a, b) or floral density (c, d) of mesocosm communities in the climate chambers of an ecotron facility. Averaged standardized values of the individual indicators contributed to averaged multifunctionality, whereas for the selected threshold, each function indicator exceeding 70% of the standardized maximum contributed to multifunctionality. Shown are means ± 95% confidence intervals. Data points are light-colored circles (n = 4).

## Discussion

Understanding the functioning of urban grassland is crucial for adaptation to climate change, the delivery of ecosystem services, and biodiversity protection in cities. Here, we assessed ecosystem functioning under climate change in mesocosms, mimicking the early establishment of road verge grasslands. Our assessment of single indicator variables and multifunctionality

emphasizes the importance of designing urban grasslands informed by ecological theory in times of climate change.

## High [$CO_2$] and temperature enhance carbon cycling during grassland establishment

RCP scenario (i.e., air temperature and [$CO_2$]) affected carbon-cycling processes and, thus, major functions for multifunctional urban grasslands. These results confirm the conclusions of other climate change experiments [66]. RCP8.5 positively affected individual functions like vegetation cover, mean height, floral abundance, and soil respiration. Therefore, attributes related to vegetation growth and habitat availability for higher trophic levels benefit from warmer and $CO_2$-richer conditions. In addition, increased soil respiration under RCP8.5 indicates an increased metabolism in the soil, accelerating C turnover driven by increased plant photosynthesis and a higher release of recently allocated C via respiration [52, 66], especially under normal water conditions.

Despite RCP8.5 favored water-use efficiency and indicator variables of productivity were greater even in water-limited mesocosm grasslands (i.e., under reduced precipitation), the interaction between RCP8.5 and normal precipitation was positively associated with higher grassland multifunctionality. For single functions and multifunctionality, elevated temperature and $CO_2$ increased multifunctionality if water supply suffices for the greater water amount required to sustain higher plant growth. This indicates warmer and $CO_2$-richer conditions favored plant development [67] due to $CO_2$ fertilization [68], particularly in young communities [36].

Since photosynthesis and respiration represent the two paramount fluxes of $CO_2$ in ecosystems linked to C allocation, increased cycling of C can be assumed from climate-change conditions such as those in RCP8.5, stimulating plant growth and soil respiration [10]. Plants under RCP8.5 assimilate more C [68] and allocate more of the photosynthesized C below-ground, leading to increased soil respiration [52] because warmer temperatures accelerate metabolic activity in the soil. While previous studies on urban lawns under standard municipal and private management (e.g., frequent mowing, mulching) showed that they have higher respiration rates and contribute to urban $CO_2$ emissions [24, 69, 70], the respiration rates recorded in our ecotron experiment were in all cases lower than those reported above (see [24]). This observation suggests that management measures rather than grassland composition and higher environmental $CO_2$ and temperature are responsible for the enhancement of $CO_2$ efflux in urban grasslands [71]. Thus, we suggest that management-related drivers of exacerbated soil respiration (e.g., frequent mowing, fertilization, and irrigation) can be a critical factor for the expected role of urban grasslands in climate-change mitigation in urban areas. Nonetheless, we acknowledge that specific urban factors, such as the high degree of surface sealing and the urban heat island effect, could play a central role in increasing $CO_2$ emissions from urban soils by up to 25% [72].

## Detrimental role of reduced precipitation on urban grassland functioning

Precipitation was a consistent driver of grassland multifunctionality across the two approaches assessed. As predicted for the growing season in Central Europe [2], lower precipitation negatively affected all functions related to carbon cycling and ecosystem multifunctionality. Reduced precipitation has more detrimental effects on carbon fixation in plant tissues (shoots, flowers, roots) than in ecosystem respiration, thus reducing net C uptake in ecosystems [52]. Previous studies also reported strong effects of rain reduction on grassland multifunctionality [12], underlining the challenging influence of modified precipitation patterns, especially in the

aboveground productivity of ecosystems [66]. In urban areas, the effects of reduced precipita-
tion might be enhanced because soil sealing and building materials intensify water loss and
stress [6]. Despite previous studies reporting the stabilizing effects of increased species richness
and adapted composition on grassland responses to drought [12, 33, 35], our findings did not
support biotic control for multifunctionality under reduced precipitation. Moreover, decreased
precipitation was detrimental for averaged multifunctionality under RCP8.5, likely because the
water-saving effect of elevated [$CO_2$] did not compensate for the temperature-driven drying of
the soil that negatively affected water supply for plant metabolism and grassland functioning.

With reduced precipitation, water retention capacity in the grasslands increased, while RCP
scenario modulated this effect. Water-stressed grasslands used more water and generated
more space in the soil and plant tissues for water retention between rain events, similar to
more engineered urban ecosystems (e.g., green roofs; [53]). Notwithstanding, less water reten-
tion occurred under RCP8.5, given the increased efficiency in water use by plants with elevated
[$CO_2$], preventing high water absorption by plants and reducing soil water retention capacity
following a rain event. In contrast, reduced precipitation also decreased water circulation via
ET, which supports transpirational cooling in cities [73, 74]. Particularly under RCP2.6, we
found a steeper decrease in water loss via ET, because plants tended to keep absorbed water to
sustain their metabolism. Instead, in communities growing under climate change conditions
(RCP8.5), losses in water did not substantially change due to $CO_2$ fertilization that increases
water use efficiency by plants.

## High evenness in functional type composition is beneficial for multifunctionality but not for all functions

The functional composition of the mesocosm grasslands (i.e., the proportion of forbs and
grasses) was a modulating driver of multifunctionality. Under RCP8.5, communities with an
even ratio of plant functional types (F50) exhibited higher average multifunctionality. They
favored some individual indicator variables compared to other compositions (e.g., below-
ground biomass and plant height and cover). This finding underscores the biotic control of
multifunctionality under climate change and supports the complementarity between plant
functional types as one aspect explaining the improved functioning of grasslands [32]. There-
fore, differences in composition among grasses and forbs may better explain multifunctional-
ity during the early assembly of urban grasslands, reinforcing the need to account for
functional types when assessing ecosystem function responses to climate change [34, 35].

Nevertheless, some functions were strongly correlated with the dominance of either forbs
or grasses, indicating their prominent role in specific ecosystem processes. Belowground bio-
mass was higher in grass-only communities and decreased steadily with a lower abundance of
grasses in the mixtures. The higher ability of grasses to allocate biomass to the roots and pro-
duce an intricate network to capture and accumulate resources, especially in the upper soil lay-
ers, confirms [23]. Similarly, grasses also favored vegetation cover, highlighting their ability to
cover bare soil rapidly [21] and support surface cooling [73]. In contrast, forbs increased
above-ground biomass, vegetation height, and floral abundance, significantly impacting car-
bon fixation and habitat and resources for animals, especially arthropods [25]. This contrast in
functions represents trade-offs that might reflect the degree of multifunctionality expected
from urban grasslands. Tall-grass meadows might maximize habitat and aesthetics [22, 25];
grass-dominated ones might, in turn, optimize functioning as runoff controllers [74], soil sta-
bilizers, and recreation areas [23]. Such trade-offs among functions might explain why our
averaged multifunctionality assessment underscores that high evenness between grasses and
forbs (F50) increases grassland multifunctionality under climate change.

We did not observe differential effects of functional composition on water regulation. Since urban lawns increase water capture and infiltration because their root system enables extensive soil exploration [74, 75] and the opening of channels [76], we anticipated grass-only (F0) communities to retain more water. Despite not being statistically different, communities with even functional composition (F50) retained greater amounts of water. Presumably, a high complementarity and evenness in the communities' above- and below-ground traits can benefit water capture and infiltration [77]. Furthermore, we observed the lowest water losses in grass-only communities because grasses efficiently withstand water stress [76]. In contrast, communities with even functional composition (F50) lost more water via ET and thereby contributed to water circulation in the ecosystem, one desired function of green urban areas where runoff due to sealing is high [74]. Despite water regulation not being controlled by functional composition in our experiment, evenly composed communities (F50) tended to show a greater capacity to capture water and circulate it via ET in urban grasslands.

Since many studies demonstrate that multifunctionality increases with plant diversity [31, 50, 59, 78], we did not address plant diversity here: First, we were not interested in understanding diversity effects explicitly. We sowed five (F0), 26 (F100), or 31 (F50, F75) species in our mesocosm grasslands, exemplifying real-life mixture diversity used in urban grasslands and not representing an experimentally intended gradient of plant species diversity. Second, we did not aim to disentangle species-richness influences from functional-composition ones but to understand grassland functioning in response to climate change as mediated by its functional composition. However, we cannot rule out the confounding effects of species richness and functional composition in our experimental setup. Nevertheless, two tested mesocosm grassland compositions that reflected the highest richness of sown species (31 plant species) contained the two functional types of interest and revealed contrasting multifunctionality. The mixture with high evenness between forbs and grasses (F50) performed best for most functions. In contrast, F75, also containing 31 sown species, showed lower levels of (multi)functionality. This suggests that besides diversity and functional type richness, the dominance or presence of specific functional types strongly influences ecosystem multifunctionality [79] and that in addition to plant species richness, the effects of functional composition and individual species on ecosystem functioning, e.g., biomass production [32, 78, 79] needs to be considered.

A strength of our experiment was the highly controlled factors mimicking the limited available soil volume and the strong edge effect experienced in road verge grasslands. Even though a complete set of factors related to global change could not be realized, our full-factorial combination of projected changes helps understand ecosystem responses to climate change [80]. However, we acknowledge that mesocosm experiments represent a trade-off with external validity of responses given missing realistic conditions [80], such as disturbances and interactions experienced in natural systems. Moreover, we cannot extrapolate long-term responses of grassland multifunctionality under climate change. For example, biomass production responses to $CO_2$ fertilization may stabilize over time [36, 80, 81], whereas we explicitly provide evidence of responses of recently established road verge grasslands to climate change.

The importance of the drivers of grassland multifunctionality differed between the two approaches applied in our assessment, likely due to the metric used for its evaluation [30]. The averaging approach depended mainly on the functions with large absolute values responding to the factors tested [31]. For instance, ecosystem productivity was controlled by the functional composition of the communities (i.e., forb proportion), and a similar control was found in averaged multifunctionality. In turn, we considered the 70% threshold as a reasonable assumption desired for the multifunctionality of urban grasslands, even though additional challenges and disturbances faced under heavily urbanized settings may imply the consideration of lower

thresholds of multifunctionality is necessary. In fact, including different threshold levels when assessing the multifunctionality of urban grasslands might inform management and restoration decisions due to providing an integrative assessment of the functioning of these systems. Overall, with the two approaches to multifunctionality deployed, we found high consistency in the adverse effects of reduced precipitation on the functioning of recently established road verge grasslands. The assessment of multifunctionality in restored urban grasslands and possible derived ecosystem services should be encouraged to increase their ecological value in cities.

## Conclusions

The study helps understand road verge grasslands' functioning under climate change conditions. They highlight that reduced precipitation is detrimental to the functioning of urban grasslands. At least during the early stages of their development, grasslands can benefit from warmer and $CO_2$-richer conditions that foster productivity. Moreover, our averaged multifunctionality assessment underscores the role of the functional composition of urban grasslands in modulating the responses to a changing environment. At the same time, the threshold level considered provides a realistic approach to multifunctionality. Urban grasslands with a high evenness between grasses and forbs are suitable for obtaining higher levels of multifunctionality and are more effective in biodiversity protection and ecosystem services in times of climate change.

## Supporting information

**S1 File. Description of study system.**
(DOCX)

**S2 File. Multifunctionalty at 50% threshold of multifunctionality.**
(DOCX)

**S1 Fig. Walk-in climate chambers of TUMmesa used for the mesocosm experiment with grasslands.**
(DOCX)

**S2 Fig. Realized daily air temperature, $CO_2$ concentration, and relative humidity in the climate chambers.**
(DOCX)

**S3 Fig. Experimental design deployed in the climate chambers.**
(DOCX)

**S4 Fig. Correlation matrix for the eight indicator variables of ecosystem functions assessed on mesocosm grasslands.**
(DOCX)

**S5 Fig. Clusters of seven indicator variables of ecosystem functions.**
(DOCX)

**S1 Table. Species used in the mesocosm experiment to simulate urban grasslands.**
(DOCX)

**S2 Table. Composition of each experimental grassland community with the sown proportion of the two functional types 'grasses' and 'forbs'.**
(DOCX)

**S3 Table. Overview of indicator variables of urban grasslands measured in the mesocosm experiment.**
(DOCX)

**S4 Table. Summary output of best models selected for describing drivers of single indicator variables of ecosystem functions of mesocosm grasslands.**
(DOCX)

**S5 Table. Multiple comparisons of forb proportion effects on single indicator variables of grassland functionality.**
(DOCX)

**S6 Table. Pairwise comparisons of forb proportion effects in interaction with climate-change related variables on plant hight of mesocosm grasslands.**
(DOCX)

**S7 Table. Output summary of best models for multifunctionality.**
(DOCX)

## Acknowledgments

Climate data (multi-annual air temperature in Munich) were provided by Deutscher Wetter-dienst (DWD). The Chair of Environmental Sensing and Modeling of the TUM kindly provided data on [$CO_2$] recorded in Munich. Roman Maier supported the setup and monitoring of the simulated climate change scenarios at TUMmesa. We are grateful to Sieglinde Sergl, Holger Paetsch, Luca Langlois, Paula Prucker, Phoebe Koppendorfer, Soizig Le Stradic, Veronika Kloska, and Yao Huang, who supported the experimental or data collection phase. The manuscript improved considerably thanks to helpful comments and suggestions from anonymous reviewers on earlier versions.

## Author Contributions

**Conceptualization:** Sandra Rojas-Botero, Leonardo H. Teixeira, Johannes Kollmann.

**Data curation:** Sandra Rojas-Botero.

**Formal analysis:** Sandra Rojas-Botero, Leonardo H. Teixeira.

**Investigation:** Sandra Rojas-Botero, Leonardo H. Teixeira.

**Methodology:** Sandra Rojas-Botero, Leonardo H. Teixeira, Johannes Kollmann.

**Supervision:** Johannes Kollmann.

**Visualization:** Sandra Rojas-Botero.

**Writing – original draft:** Sandra Rojas-Botero.

**Writing – review & editing:** Leonardo H. Teixeira, Johannes Kollmann.

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
