## [Decision Letter · Decision Letter 0]

26 Oct 2022

PONE-D-22-24947Low precipitation due to climate change consistently reduces multifunctionality of urban grasslands in mesocosmsPLOS ONE

Dear Dr. Rojas-Botero,

Thank you for submitting your manuscript to PLOS ONE. After careful consideration, we feel that it has merit but does not fully meet PLOS ONE’s publication criteria as it currently stands. Therefore, we invite you to submit a revised version of the manuscript that addresses the points raised during the review process.

We look forward to receiving your revised manuscript.

Kind regards,

Xiao-Dong Yang

Academic Editor

PLOS ONE

Journal Requirements:

Reviewers' comments:

Reviewer's Responses to Questions

**Comments to the Author**

1. Is the manuscript technically sound, and do the data support the conclusions?

Reviewer #1: Yes

Reviewer #2: Yes

2. Has the statistical analysis been performed appropriately and rigorously? 

Reviewer #1: Yes

Reviewer #2: Yes

3. Have the authors made all data underlying the findings in their manuscript fully available?

Reviewer #1: Yes

Reviewer #2: No

4. Is the manuscript presented in an intelligible fashion and written in standard English?

Reviewer #1: Yes

Reviewer #2: Yes

5. Review Comments to the Author

Reviewer #1: This manuscript is a research on the multi-function of urban ecosystem, which has important research significance. But I still have the following questions and questions to be answered by the author

1.In the experimental design, what is the basis for the four mixing ratios of F0, F50, F75 and F100? Whether the distribution of the density of different grass species of F50 and F75 during mixed sowing is determined based on the previous community survey on the grassland at the edge of multiple urban roads？

2. Why should RCP8.5 be chosen as the worst scenario? This extreme situation is unlikely to occur. What is the significance of choosing it to study the urban grassland ecosystem? Why not consider RCP6.0 and RCP4.5?

3. The results in line 237 show that the communities with weeds have more aboveground biomass in different environments than the local grass communities. Will the growth periods of local grass and weeds show different trends in a certain period? For example, because of climate change, the growth period is extended, leading to an increase in biomass; The biomass is reduced due to the premature maturity.

4.There are some misunderstandings about the results of plant height and coverage. For example, under different environmental conditions, will the same proportion of different species of grass cause the overall plant community height to be higher due to competition or other relationships? If so, is this phenomenon worth considering?

5. How is water retention reflected? Is it the water retention and loss of plants during transpiration or the water retention of soil?

Reviewer #2: Urban grasslands are crucial for biodiversity and ecosystem services in cities. This study studied the effects of simulated climate change, temperature rise and precipitation decrease on the individual function and overall multi-function of the grassland in the Middle World. The results show that grassland with uniform proportion of plant function types can better cope with climate change and is a good choice to increase the benefits of urban green infrastructure. This study provides experimental evidence for the impact of climate change on urban ecosystem functions. Designing the composition of urban grassland based on ecological theory may increase its adaptability to global change. However, there are still many problems in the current manuscript.

The specific issues are as follows:

1. Introduction：How to consider the impact of human factors on urban grassland? Although urban grassland is crucial to urban biodiversity and ecosystem services, it is also inseparable from human factors. So please add relevant content.

2. Materials and methods：(1) Which herbaceous or seeds do these four kinds of grassland mixtures mainly include? I am very confused. (2) Line 158: What is the mean precipitation in 2000 and 2019? How is this controlled? Need to elaborate. (3) Line 167-169: When are the five variables related to vegetation recorded (i.e. aboveground and underground biomass, flower yield, plant cover and plant height)? (4) Line 172: What are the methods or steps for measuring soil respiration?

3. Date analysis: (1) Line 208-210: In the threshold method, why is the 70% threshold selected as the ideal level of multi-functional grassland on the edge of urban roads?

4. Discussion：(1) Line 356-361: Only the observation results show that management measures, rather than grassland composition and climate change conditions, are related to the disproportionate increase of carbon dioxide emissions from urban grassland, which obviously needs more verification. Because, this part is related to human factors, but the full text has never considered these factors.

6. Other：(1) The resolution of Figure 1 is not high, and the picture is blurry. Besides, control experiment and outdoor conditions need to be marked in the picture.

In a word, this study has good innovation and scientific significance. However, there are still some problems to be noted.

6. PLOS authors have the option to publish the peer review history of their article (what does this mean?). If published, this will include your full peer review and any attached files.

Reviewer #1: No

Reviewer #2: No

---

## [Author Response · Author response to Decision Letter 0]

15 Dec 2022

Dr. Xiao-Dong Yang 

Academic Editor

PLOS ONE

Freising, 04.11.2022

Dear Dr. Yang,

Thank you for the helpful feedback on our manuscript entitled Low precipitation due to climate change consistently reduces multifunctionality of mesocosm urban grasslands (PONE-D-22-24947). We appreciate the opportunity to revise and improve it based on the comments of the two reviewers. Herewith we submit the revised version of the manuscript with the required structure and more clearly stated information, mainly reflecting more detailed aspects presented in the method section.

In the following, we respond to each of the points raised by the reviewers:

Reviewer #1: This manuscript is a research on the multi-function of urban ecosystem, which has important research significance. But I still have the following questions and questions to be answered by the author

1. In the experimental design, what is the basis for the four mixing ratios of F0, F50, F75 and F100? Whether the distribution of the density of different grass species of F50 and F75 during mixed sowing is determined based on the previous community survey on the grassland at the edge of multiple urban roads？

Thanks for raising this point that now can be found more explicitly in the manuscript (L 111-130). We were interested in testing an array of grassland compositions suitable for urban grasslands, specifically road verges, using existing knowledge on the proportions between forbs and grasses in similar grassland communities and our practical experience. We selected the two extremes of the possible ratios, i.e., F0 (only grasses) and F100 (only forbs), to test expected opposite responses in multifunctionality resulting from the absence of one of the two functional groups investigated. Additionally, previous research on seed mixtures for restoring road verges found that the percentage of grass biomass, which correlates to grass seed content in mixtures, had a significant negative effect on the number of target species established (Staab et al. 2015). 

Since a higher percentage of grass biomass leads to a decrease in species number above a 'threshold' of 70% grass coverage, we considered that a mixture forb:grass above 50:50 (e.g., 25:75) would substantially drop forb-related functions of interest in our tested communities and reduce target forb species of the community. Additionally, we expect to observe linear responses in grassland communities (in terms of species composition and functionality) until having a very high proportion of grasses in relation to other functional types (i.e., 70% grasses). Above this 'threshold,' non-linear responses should be expected. Thus, the comparisons of proportions above such level (e.g., >80% grasses) would, most likely, not add any new meaningful explanation to the observed patterns and relationships (i.e., produce non-significant results). Therefore, as a compromise regarding the responses obtained and resources or space available to test new compositions, we decided to focus on those that would produce more interesting results for designing multifunctional urban grasslands. Furthermore, from the practical side, we know that standard or commercial urban lawns are designed based on mixtures strictly composed of grasses. In contrast, increasing calls to promote the diversity of wildflowers and associated resources for urban pollinators have fostered the preparation and commercialization of forb mixtures to be widely distributed in urban areas. Additionally, considering the feasibility of establishing 'pollinator-friendly' meadows, the common practice favors the utilization of mixtures similar to our F50 composition (e.g., https://www.rieger-hofmann.de/sortiment-shop/mischungen/begruenungen-fuer-den-stadt-und-siedlungsbereich/14-verkehrsinselmischung.html). Finally, it is important to point out that our experimental restoration sites established in actual urban road verges were sown with only one seed mixture type (F100) and, after three years of assessment, present compositions of 26.1 ± 20.7% grasses (unpublished data), thus reflecting the uneven proportion tested in our mesocosm experiment (F75).

2. Why should RCP8.5 be chosen as the worst scenario? This extreme situation is unlikely to occur. What is the significance of choosing it to study the urban grassland ecosystem? Why not consider RCP6.0 and RCP4.5?

We complemented our explanation of this choice (L 145-150). Even though the likelihood of the different scenarios is debated in the scientific community and ongoing research attempts to define new scenarios based on the most recent data, we used the RCP scenarios delivered by IPCC AR5 (IPCC 2014). (i) We were interested in evaluating clearly contrasting cases of environmental conditions related to climate change and their effects on urban grassland multifunctionality. Thus, we kept to IPCC terminology of RCP8.5 as often-stated 'worst case' in the literature. (ii) Since our model system is planned for urban areas, where the effects of climate change are expected to be enhanced by phenomena such as heat islands and a high discharge of CO2 to the atmosphere (Masson et al. 2020; Zhao et al. 2021), we considered RCP8.5 a realistic extreme in an urban context. Finally, (iii) current research states, "Not only are the emissions consistent with RCP8.5 in close agreement with historical total cumulative CO2 emissions (within 1%), but RCP8.5 is also the best match out to mid-century under current and stated policies with still highly plausible levels of CO2 emissions in 2100," (Schwalm et al. 2020). Nonetheless, we agree that it might be worth investigating the effects of other climate change scenarios and other environmental factors, such as N-deposition and soil contamination by petroleum hydrocarbons and other pollutants, on the multifunctionality of urban ecosystems. However, more sophisticated designs are needed for such studies because our results show that some responses are inconclusive. Disentangling more effects requires a more detailed experimental approach and an extended assessment period.

3. The results in line 237 show that the communities with weeds have more aboveground biomass in different environments than the local grass communities. Will the growth periods of local grass and weeds show different trends in a certain period? For example, because of climate change, the growth period is extended, leading to an increase in biomass; The biomass is reduced due to the premature maturity. 

If we understand correctly the point raised by the reviewer, we would like to inform that we expected a linear increase in the communities' biomass during the experiment. Additionally, in case the reviewer's comment addresses a potential comparison between grass-only and forb-only communities, we argue that both community types presented similar responses in terms of biomass production over time, with forb-only communities showing a steeper curve (i.e., faster biomass increase over time). As stated in the manuscript, forb-dominated communities produced more aboveground biomass than grass-dominated communities did, while we measured this biomass production only at the end of the experiment that lasted ten weeks. Still, we tend to agree with the rationale that climate change would speed up plant development because a longer development time and slower growth could be expected from plants growing under RCP2.6. In turn, plants growing under RCP8.5 would have a shorter development period and exhibit faster growth rates. However, since we cannot know or could not measure this due to the time extension of our experiment, it would be no more than speculation to imply changes in biomass production at later points of the vegetation period, which could have occurred.

4. There are some misunderstandings about the results of plant height and coverage. For example, under different environmental conditions, will the same proportion of different species of grass cause the overall plant community height to be higher due to competition or other relationships? If so, is this phenomenon worth considering? 

Thanks for raising this interesting point. The composition F50 increased the ratio between community height and cover in response to CO2 fertilization. Since our experiment did not last enough for the communities' development to be limited by soil nutrient availability, a surplus of C favored size increase in communities evenly composed of forbs and grasses (i.e., F50), even under reduced water levels, in comparison to the other compositions (F0, F75, and F100). We argue that rather than a competition effect on height and cover as suggested by the reviewer, this pattern underscores potential complementarity effects in resource use resulting from a more balanced distribution of plant functional characteristics in F50 communities, thus resulting in an even occupation of ecological niches available. With even fewer limitations to plant growth (i.e., under normal precipitation scenario), F50 was also very effective in increasing the ratio between community height and cover (Fig. 1), depicting a consistent increase in the growth ratio mainly due to environmental factors. Additionally, as explained before in response to the 3rd point raised by the reviewer, we can confirm that plant height was greater due to more forbs in the communities when comparing both RCP scenarios (RCP2.6 and 8.5). Differently, plant cover was correspondingly greater with increasing the proportion of grasses in the communities.

Fig. 1 Change in the ratio height:cover (mean ± SE) in mesocosm grasslands grown for ten weeks under simulated climate change scenarios (RCP2.6 and 8.5) and two levels of mean seasonal precipitation (reduced [-50% rain], and normal). Different functional compositions based on the proportion forbs:grasses were tested to better understand climate change effects on different ecosystem functions. A functional composition with maximum evenness forb:grass (F50; blue lines) profited the most from CO2 fertilization compared to the other compositions, even under water scarcity. 

5. How is water retention reflected? Is it the water retention and loss of plants during transpiration or the water retention of soil?

In this case, we would like to refer to the information reported in detail in the supplementary material. The method we applied for measuring water retention is based on a heavy rain event simulation by watering from above with a spraying hose. After the simulated rainfall, we measured: (a) the fraction of water retained (both in soil and above- and belowground plant structures) after one hour of completing the rainfall simulation, and (b) the fraction of evapotranspired water after 24 hours of the rainfall, using a gravimetric method. We emphasized that 'water retention' refers to the fraction of retained water in the soil and the plant communities, i.e., the plant-soil system (L 197-200). The references for the method used in other urban green infrastructures (MacIvor & Lundholm 2011; MacIvor et al. 2018) are included. We stated that more detailed information is available in the supplementary material (L 188-190). 

Reviewer #2: Urban grasslands are crucial for biodiversity and ecosystem services in cities. This study studied the effects of simulated climate change, temperature rise and precipitation decrease on the individual function and overall multi-function of the grassland in the Middle World. The results show that grassland with uniform proportion of plant function types can better cope with climate change and is a good choice to increase the benefits of urban green infrastructure. This study provides experimental evidence for the impact of climate change on urban ecosystem functions. Designing the composition of urban grassland based on ecological theory may increase its adaptability to global change. However, there are still many problems in the current manuscript. 

The specific issues are as follows:

1. Introduction: How to consider the impact of human factors on urban grassland? Although urban grassland is crucial to urban biodiversity and ecosystem services, it is also inseparable from human factors. So please add relevant content.

We appreciate the feedback and the critical comments to improve the manuscript. In the introduction, we highlight that urban grasslands are frequent in cities while strongly influenced by human management (e.g., mowing and fertilization) and increasingly susceptible to restoration and implementation as novel ecosystems. Striving for a concise introduction, we directly addressed the environmental factors of our interest in climate change. We did not include other human-driven disturbances such as trampling, artificial light, pollution, etc. Nonetheless, we added (L 56-63) brief information on the typical 'novel' character of urban grasslands and the additional disturbances encountered in urban grasslands that may affect their composition and functioning. Despite being ecosystems with a substantial degree of human intervention in cities and a designed character, urban grasslands are still a fair object of scientific endeavor. Furthermore, in the related supplementary material S1 File, we describe the characteristics of the specific model system (urban roadside grasslands) and the diversity of human-related challenges for these ecosystems.

2. Materials and methods: (1) Which herbaceous or seeds do these four kinds of grassland mixtures mainly include? I am very confused. (2) Line 158: What is the mean precipitation in 2000 and 2019? How is this controlled? Need to elaborate. (3) Line 167-169: When are the five variables related to vegetation recorded (i.e. aboveground and underground biomass, flower yield, plant cover and plant height)? (4) Line 172: What are the methods or steps for measuring soil respiration?

Concerning the questions arising from the methods, most of the requested information is detailed in the supplementary material to avoid an overwhelming material and methods section. Information on species and respective proportions in the mixtures are provided in the S1 Table and S2 Table. An expanded explanation of the selected functions and dates for measuring each indicator variable is in the S3 Table. A more conspicuous reference to this material and contained information was added in the method section (L187-190). Furthermore, we included extra information in the supplementary material detailing information on the 'ecotron' research facility TUMmesa. We added additional information on the source for the freely available data on precipitation recorded in the weather station of Munich.

3. Data analysis: (1) Line 208-210: In the threshold method, why is the 70% threshold selected as the ideal level of multi-functional grassland on the edge of urban roads?

We appreciate your interest in this criterion. We are aware that grasslands established in urban road verges are subjected to many challenging conditions that constrain an ambitious level of multifunctionality. We based our choice on the current available literature. For example, Lozano et al. (2021) studied multifunctionality at a threshold of 70% in soil subjected to drought and microplastic addition (common disturbances in urban ecosystems). They found this threshold was not overambitious compared to 30 and 50%. Similarly, Strobl et al. (2019) found that multiple functions did not simultaneously reach an equally desirable level of functioning (i.e., threshold >80%) in restored peatlands, suggesting thresholds above 80% may be over-optimistic in restored ecosystems in the short-midterm. Moreover, as van der Plas et al. (2016) stated, a threshold of multifunctionality of 70% is realistic and responsive to ecosystem management (in their case, species addition). All in all, we decided to keep a medium-high ambitioning level of multifunctionality to test in our experimental grasslands. We also expanded this information in L 227-232

4. Discussion：(1) Line 356-361: Only the observation results show that management measures, rather than grassland composition and climate change conditions, are related to the disproportionate increase of carbon dioxide emissions from urban grassland, which obviously needs more verification. Because, this part is related to human factors, but the full text has never considered these factors.

We rephrase the info to make it more straightforward. We fully acknowledge we did not measure any of the mentioned factors. Instead, we use findings in the literature pointing out which drivers of disproportionate respiration of urban grasslands have been recorded. We use, e.g., Decina (2016), Townsend-Small & Czimczik (2010), Lerman & Contosta (2019), and Hill et al. (2021), that present high emissions recorded in urban lawns due to management standards of urban grasslands. It now reads: " While previous studies on urban lawns under standard municipal and private management (e.g., frequent mowing, mulching) showed that they have higher respiration rates and contribute to urban CO2 emissions [24,69,70], the respiration rates recorded in our ecotron experiment were in all cases lower than those reported above [see 24]. This observation suggests that management measures rather than grassland composition and higher environmental CO2 and temperature are responsible for enhancing CO2 efflux in urban grasslands [71]. Thus, we suggest that management-related drivers of exacerbated soil respiration (e.g., frequent mowing, fertilization, and irrigation) can be a critical factor for the expected role of urban grasslands in climate-change mitigation in urban areas. Nonetheless, we acknowledge that specific urban factors, such as the high degree of surface sealing and the urban heat island effect, could significantly increase CO2 emissions from urban soils by up to 25% [72]." (L 379-390)

5. Other: (1) The resolution of Figure 1 is not high, and the picture is blurry. Besides, control experiment and outdoor conditions need to be marked in the picture.

Thanks for pointing this out. We reworked the picture resolution that could have been affected during the online upload and detailed the suggested information. For a better overview of the experimental communities developed under the two climate change scenarios, S1 Fig was referenced.

In a word, this study has good innovation and scientific significance. However, there are still some problems to be noted.

Thanks for valuing the contribution and for the suggestions to get the manuscript to a better state!

Sincerely,

(SANDRA ROJAS BOTERO, ON BEHALF OF ALL AUTHORS)

Doctoral Candidate,

Chair of Restoration Ecology, Technical University of Munich, Germany

 

References

Decina SM, Hutyra LR, Gately CK, Getson JM, Reinmann AB, Short Gianotti AG, Templer PH (2016) Soil respiration contributes substantially to urban carbon fluxes in the greater Boston area. Environmental pollution 212:433–439. doi: 10.1016/j.envpol.2016.01.012

Hill AC, Barba J, Hom J, Vargas R (2021) Patterns and drivers of multi-annual CO2 emissions within a temperate suburban neighborhood. Biogeochemistry 152:35–50. doi: 10.1007/s10533-020-00731-1

IPCC (ed) (2014) Climate Change 2014: Synthesis Report. Contribution of Working Groups I, II and III to the Fifth Assessment Report of the Intergovernmental Panel on Climate Change. IPCC, Geneva, Switzerland

Lerman SB, Contosta AR (2019) Lawn mowing frequency and its effects on biogenic and anthropogenic carbon dioxide emissions. Landscape and Urban Planning 182:114–123. doi: 10.1016/j.landurbplan.2018.10.016

Lozano YM, Aguilar‐Trigueros CA, Onandia G, Maaß S, Zhao T, Rillig MC (2021) Effects of microplastics and drought on soil ecosystem functions and multifunctionality. Journal of Applied Ecology 58:988–996. doi: 10.1111/1365-2664.13839

MacIvor JS, Lundholm J (2011) Performance evaluation of native plants suited to extensive green roof conditions in a maritime climate. Ecological Engineering 37:407–417. doi: 10.1016/j.ecoleng.2010.10.004

MacIvor JS, Sookhan N, Arnillas CA, Bhatt A, Das S, Yasui S-LE, Xie G, Cadotte MW (2018) Manipulating plant phylogenetic diversity for green roof ecosystem service delivery. Evolutionary applications 11:2014–2024. doi: 10.1111/eva.12703

Masson V, Lemonsu A, Hidalgo J, Voogt J (2020) Urban Climates and Climate Change. Annual Review of Environment and Resources 45:411–444. doi: 10.1146/annurev-environ-012320-083623

Schwalm CR, Glendon S, Duffy PB (2020) RCP8.5 tracks cumulative CO2 emissions. Proceedings of the National Academy of Sciences of the United States of America 117:19656–19657. doi: 10.1073/pnas.2007117117

Staab K, Yannelli FA, Lang M, Kollmann J (2015) Bioengineering effectiveness of seed mixtures for road verges: Functional composition as a predictor of grassland diversity and invasion resistance. Ecological Engineering 84:104–112. doi: 10.1016/j.ecoleng.2015.07.016

Strobl K, Kollmann J, Teixeira LH (2019) Integrated assessment of ecosystem recovery using a multifunctionality approach. Ecosphere 10:95. doi: 10.1002/ecs2.2930

Townsend-Small A, Czimczik CI (2010) Carbon sequestration and greenhouse gas emissions in urban turf. Geophysical Research Letters 37:n/a-n/a. doi: 10.1029/2009GL041675

van der Plas F, Manning P, Allan E, Scherer-Lorenzen M, Verheyen K, Wirth C, Zavala MA, Hector A, Ampoorter E, Baeten L, Barbaro L, Bauhus J, Benavides R, Benneter A, Berthold F, Bonal D, Bouriaud O, Bruelheide H, Bussotti F, Carnol M, Castagneyrol B, Charbonnier Y, Coomes D, Coppi A, Bastias CC, Muhie Dawud S, Wandeler H de, Domisch T, Finér L, Gessler A, Granier A, Grossiord C, Guyot V, Hättenschwiler S, Jactel H, Jaroszewicz B, Joly F-X, Jucker T, Koricheva J, Milligan H, Müller S, Muys B, Nguyen D, Pollastrini M, Raulund-Rasmussen K, Selvi F, Stenlid J, Valladares F, Vesterdal L, Zielínski D, Fischer M (2016) Jack-of-all-trades effects drive biodiversity-ecosystem multifunctionality relationships in European forests. Nature communications 7:11109. doi: 10.1038/ncomms11109

Zhao L, Oleson K, Bou-Zeid E, Krayenhoff ES, Bray A, Zhu Q, Zheng Z, Chen C, Oppenheimer M (2021) Global multi-model projections of local urban climates. Nature Climate Change 11:152–157. doi: 10.1038/s41558-020-00958-8

---

## [Decision Letter · Decision Letter 1]

10 Jan 2023

Low precipitation due to climate change consistently reduces multifunctionality of urban grasslands in mesocosms

PONE-D-22-24947R1

Dear Dr. Rojas-Botero,

We’re pleased to inform you that your manuscript has been judged scientifically suitable for publication and will be formally accepted for publication once it meets all outstanding technical requirements.

Kind regards,

Xiao-Dong Yang

Academic Editor

PLOS ONE

Reviewers' comments:

Reviewer's Responses to Questions

**Comments to the Author**

1. If the authors have adequately addressed your comments raised in a previous round of review and you feel that this manuscript is now acceptable for publication, you may indicate that here to bypass the “Comments to the Author” section, enter your conflict of interest statement in the “Confidential to Editor” section, and submit your "Accept" recommendation.

Reviewer #1: All comments have been addressed

Reviewer #2: (No Response)

2. Is the manuscript technically sound, and do the data support the conclusions?

Reviewer #1: Yes

Reviewer #2: (No Response)

3. Has the statistical analysis been performed appropriately and rigorously? 

Reviewer #1: Yes

Reviewer #2: (No Response)

4. Have the authors made all data underlying the findings in their manuscript fully available?

Reviewer #1: Yes

Reviewer #2: (No Response)

5. Is the manuscript presented in an intelligible fashion and written in standard English?

Reviewer #1: (No Response)

Reviewer #2: (No Response)

6. Review Comments to the Author

Reviewer #1: (No Response)

Reviewer #2: (No Response)

7. PLOS authors have the option to publish the peer review history of their article (what does this mean?). If published, this will include your full peer review and any attached files.

Reviewer #1: No

Reviewer #2: No

---

## [Editor Report · Acceptance letter]

23 Jan 2023

PONE-D-22-24947R1 

Low precipitation due to climate change consistently reduces multifunctionality of urban grasslands in mesocosms 

Dear Dr. Rojas-Botero:

I'm pleased to inform you that your manuscript has been deemed suitable for publication in PLOS ONE. Congratulations! Your manuscript is now with our production department. 

Kind regards, 

on behalf of

Dr. Xiao-Dong Yang 

Academic Editor

PLOS ONE